# A Novel Mixed-Attribute Fusion-Based Naive Bayesian Classifier

Guiliang Ou[1], Yulin He [1,2,*], Philippe Fournier-Viger [1,2] and Joshua Zhexue Huang [1,2]

1   Big Data Institute, College of Computer Science and Software Engineering, Shenzhen University, Shenzhen 518060, China
2   Guangdong Laboratory of Artificial Intelligence and Digital Economy (SZ), Shenzhen 518107, China
*   Correspondence: yulinhe@gml.ac.cn; Tel.: +86-185-3131-5747

**Abstract:** The Naive Bayesian classifier (NBC) is a well-known classification model that has a simple structure, low training complexity, excellent scalability, and good classification performances. However, the NBC has two key limitations: (1) it is built upon the strong assumption that condition attributes are independent, which often does not hold in real-life, and (2) the NBC does not handle continuous attributes well. To overcome these limitations, this paper presents a novel approach for NBC construction, called mixed-attribute fusion-based NBC (MAF-NBC). It alleviates the two aforementioned limitations by relying on a mixed-attribute fusion mechanism with an improved autoencoder neural network for NBC construction. MAF-NBC transforms the original mixed attributes of a data set into a series of encoded attributes with maximum independence as a pre-processing step. To guarantee the generation of useful encoded attributes, an efficient objective function is designed to optimize the weights of the autoencoder neural network by considering both the encoding error and the attribute's dependence. A series of persuasive experiments was conducted to validate the feasibility, rationality, and effectiveness of the designed MAF-NBC approach. Results demonstrate that MAF-NBC has superior classification performance than eight state-of-the-art Bayesian algorithms, namely the discretization-based NBC (Dis-NBC), flexible naive Bayes (FNB), tree-augmented naive (TAN) Bayes, averaged one-dependent estimator (AODE), hidden naive Bayes (HNB), deep feature weighting for NBC (DFW-NBC), correlation-based feature weighting filter for NBC (CFW-NBC), and independent component analysis-based NBC (ICA-NBC).

**Keywords:** naive Bayesian classifier; attribute independence assumption; mixed-attribute classification; conditional probability; Bayesian network; attribute transformation





## 1. Introduction

As one of the top 10 algorithms in the fields of data mining and machine learning [1], the naive Bayesian classifier (NBC) has been used in numerous domains. The main advantage of the NBC is its simple model structure that makes it easy to implement and its good theoretical interpretability. In recent years, the NBC also received much attention from the industry and academia since it can be easily deployed in distributed environments to process big data. Despite possessing several desirable properties, the NBC is built upon a strong assumption, called the attribute independence assumption, which states that condition attributes must be mutually independent with respect to the decision attribute. This assumption simplifies the calculation of posterior probabilities (the probabilities that attribute values of a sample are all observed for a given class). Rather than computing a posterior probability as a joint probability of condition attributes, the NBC calculates it as the product of multiple marginal probabilities. This makes the computation of NBC very efficient and allows the estimation of probabilities even with small data sets. However, the attribute independence assumption does not hold in many real-life data sets, which substantially limits the prediction performance of the NBC. Recent studies on the NBC mainly focused on finding ways to relax the independence assumption so as to further improve its generalization performance. There are two main approaches for improving the

testing ability of the NBC [2]: (1) improving the structure of models and (2) performing data transformations. The former approach introduces complex Bayesian network structures to model attribute dependencies, while the latter applies feature selection or feature extraction methods to select independent attributes. Some representative studies of these two approaches are described next.

Model structure-oriented improvement methods use different estimation strategies (e.g., density estimation, Bayesian network, and attribute weighting) to estimate the conditional probability that a sample's attribute values are observed given its class. The flexible naive Bayes (FNB) [3] relies on kernel density estimations to determine a class's conditional probabilities. However, FNB can only handle continuous attributes and does not offer a solution to relax the attribute independence assumption. The tree-augmented naive (TAN) Bayes [4] algorithm is a semi-naïve Bayesian learning method that relaxes the attribute independence assumption by employing a tree structure, where each condition attribute only depends on the decision attribute and at most one condition attribute. The averaged one-dependent estimator (AODE) [5] is an ensemble classifier that applies an attribute selection algorithm to construct a series of one-dependent classifiers. Each classifier is a simple Bayesian network that is obtained by averaging all one-dependence classifiers. The hidden naive Bayes (HNB) classifier [6] uses the weighted sum of two-attribute dependencies to represent multiple-attribute dependence, where the weights are determined based on the mutual information between two condition attributes. The deep feature weighting for NBC (DFW-NBC) [7] technique and correlation-based feature weighting filter for NBC (CFW-NBC) [8] are two attribute weighting-based NBCs. DFW-NBC estimates each conditional probability of the NBC by deeply computing attribute weighted frequencies from training data while CFW-NBC weights the condition attributes by considering both the mutual relevance and the average mutual redundancy.

Data transformation-oriented improvement methods focus on the deep exploration of the training data to avoid complex and time-consuming structure learning. The NBCs obtained by this approach still rely on the independence assumption; thus, the performance may be degraded if the training data do not satisfy that assumption. Feature extraction techniques are the most commonly used to transform data with attribute dependencies into data that have no dependencies. Bressan and Vitria [9] applied class-conditional independent component analysis (CC-ICA) to improve the classification performance of the NBC. Qin et al. [10] proposed an ICA-based NBC (ICA-NBC) to improve the classification performance of the NBC. Fan and Poh [11] evaluated the classification performance of the NBC built using three feature extraction methods, namely principal component analysis (PCA), ICA, and CC-ICA. Experimental results have shown that PCA, ICA, and CC-ICA can slightly increase the testing accuracies of the NBC on the selected data sets. Jayanthi and Sasikala [12] trained an NBC based on website attributes extracted by PCA to perform web link spam detection. Zhang et al. [13] applied PCA to extract key attributes in network data and then trained the NBC to conduct network intrusion detection.

Although the aforementioned methods can improve the classification performance of the NBC for specific application scenarios, these methods do not provide a good solution for handling mixed attributes (continuous and categorical attributes). Some studies [14,15] revealed that the discretization of continuous attributes can lead to losing precious information about the original data set, while the determination of an optimal Bayesian network structure is an NP-hard problem [16]. To address the above limitations, this paper proposes a new strategy to enhance the generalization capability of the NBC for the mixed-attribute classification problem. The proposed approach not only retains as much information as possible about the original data but also keeps the model structure as simple as possible. The contributions of this paper are summarized as follows. A new mixed attribute fusion-based NBC (MAF-NBC) is proposed for mixed-attribute classification problems. To relax the attribute independence assumption, an autoencoder neural network (ANN) based on the minimization of both encoding error and attribute dependence is iteratively trained to transform the original mixed attributes into a series of independent and encoded at-

tributes. Extensive experiments conducted on multiple data sets demonstrate the feasibility, rationality, and effectiveness of the MAF-NBC.

The remainder of this paper is organized as follows. Section 2 reviews the principles of the NBC. Section 3 presents the proposed MAF-NBC. Section 4 reports results from experiments to assess the performance of MAF-NBC. Finally, Section 5 concludes this paper and describes future studies.

## 2. Naive Bayesian Classifier for Mixed-Attribute Classification

Let there be a classification data set $\mathbb{D} = \bigcup\limits_{m=1}^{\mathcal{M}} \mathbb{D}^{(m)}$ containing $\mathcal{N}$ samples. Each sample has $\mathcal{D}$ condition attributes including $\mathcal{D}_1$ discrete attributes and $\mathcal{D}_2$ continuous attributes. Moreover, all samples are divided into $\mathcal{M}$ different classes, where $\mathcal{N}_m$ denotes the number of samples that belong to the $m$-th ($m = 1, 2, \cdots, \mathcal{M}$) class.

$$
\mathbb{D}^{(m)} = \left\{
\begin{array}{r}
\left( \mathrm{x}_n^{(m)}, y_n^{(m)} \right) \Big| \mathrm{x}_n^{(m)} = \left( a_{n1}^{(m)}, \cdots, a_{n\mathcal{D}_1}^{(m)}, b_{n1}^{(m)}, \cdots, b_{n\mathcal{D}_2}^{(m)} \right), \\
y_n^{(m)} = w_m, n = 1, 2, \cdots, \mathcal{N}_m
\end{array}
\right\},
\tag{1}
$$

$\sum\limits_{m=1}^{\mathcal{M}} \mathcal{N}_m = \mathcal{N}, \mathcal{D}_1 + \mathcal{D}_2 = \mathcal{D}, a_{ni}^{(m)}, i \in \{1, 2, \cdots, \mathcal{D}_1\}$ is the $i$-th discrete attribute value, $b_{nj}^{(m)}, j \in \{1, 2, \cdots, \mathcal{D}_2\}$ is the $j$-th continuous attribute value, and $\{w_1, w_2, \cdots, w_{\mathcal{M}}\}$ is the class label set. The next paragraphs explain the principles of the naive Bayesian classifier (NBC) for the mixed-attribute classification problem on data set $\mathbb{D}$.

Assume that there is a new sample $\mathrm{x} = \left( a_1, \cdots, a_{\mathcal{D}_1}, b_1, \cdots, b_{\mathcal{D}_2} \right)$. The NBC determines its class label by using this equation:

$$
\begin{aligned}
y &= \underset{m=1,2,\cdots,\mathcal{M}}{\arg\max} \ \mathrm{P}(w_m | \mathrm{x}) \\
&= \underset{m=1,2,\cdots,\mathcal{M}}{\arg\max} \ \frac{\mathrm{P}(\mathrm{x}|w_m)\mathrm{P}(w_m)}{\mathrm{P}(\mathrm{x})}, \\
&\propto \underset{m=1,2,\cdots,\mathcal{M}}{\arg\max} \ \mathrm{P}(\mathrm{x}|w_m)\mathrm{P}(w_m)
\end{aligned}
\tag{2}
$$

where $\mathrm{P}(w_m|\mathrm{x})$ is the posterior probability, $\mathrm{P}(w_m)$ is the prior probability, and $\mathrm{P}(\mathrm{x}|w_m)$ is the conditional probability. Generally, the prior probability in Equation (1) can be calculated as follows.

$$
\mathrm{P}(w_m) = \frac{\mathcal{N}_m}{\mathcal{N}}.
\tag{3}
$$

The key of training an NBC for the mixed-attribute classification problem is to calculate the conditional probability based on the independent attribute assumption as follows.

$$
\begin{aligned}
\mathrm{P}(\mathrm{x}|w_m) &= \mathrm{P}\left( a_1, \cdots, a_{\mathcal{D}_1}, b_1, \cdots, b_{\mathcal{D}_2} | w_m \right) \\
&= \left[ \prod_{i=1}^{\mathcal{D}_1} \mathrm{P}(a_i | w_m) \right] \left[ \prod_{j=1}^{\mathcal{D}_2} \mathrm{P}(b_j | w_m) \right].
\end{aligned}
\tag{4}
$$

Conditional probability $\mathrm{P}(a_i | w_m)$ corresponding to a discrete attribute value $a_i$ is calculated as follows:

$$
\mathrm{P}(a_i | w_m) = \frac{\sum\limits_{n=1}^{\mathcal{N}_m} \mathrm{I}\left( a_i, a_{ni}^{(m)} \right)}{\mathcal{N}_m},
\tag{5}
$$

where

$$
\mathrm{I}(u, v) = \left\{
\begin{array}{ll}
1, & \text{if } u = v \\
0, & \text{if } u \neq v
\end{array}
\right.
\tag{6}
$$

is an indicator function that is used to count the frequency of $a_i$ in the $i$-th continuous attribute values $a_{1i}^{(m)}, a_{2i}^{(m)}, \cdots, a_{\mathcal{N}_{m,j}}^{(m)}$ corresponding to samples belonging to the $m$-th class. John and Langley [3] constructed a flexible NBC (FNBC), which used the kernel density estimation technique [17] to calculate the term of $P(b_j|w_m)$ for the continuous attribute value in Equation (3) as follows:

$$P(b_j|w_m) \propto p(b_j|w_m) = \frac{1}{\mathcal{N}_m} \sum_{n=1}^{\mathcal{N}_m} \frac{1}{\sqrt{2\pi}h_j} \exp\left[ -\frac{1}{2}\left( \frac{b_j - b_{nj}^{(m)}}{h_j} \right)^2 \right] \tag{7}$$

where $p(b_j|w_m)$ is the estimated probability density function (*p.d.f.*) value of $b_j$ based on the $j$-th continuous attribute values $b_{1j}^{(m)}, b_{2j}^{(m)}, \cdots, b_{\mathcal{N}_{m,j}}^{(m)}$ corresponding to samples from the $m$-th class, and $h_j > 0$ ($h_j = \frac{1}{\sqrt{\mathcal{N}_m}}$ in [3]) is the bandwidth parameter. In addition, continuous attribute discretization can also be used to determine $p(b_j|w_m)$ as follows:

$$P(b_j|w_m) = P(c_j|w_m) = \frac{\sum\limits_{n=1}^{\mathcal{N}_m} I\left(c_j, c_{nj}^{(m)}\right)}{\mathcal{N}_m} \tag{8}$$

by transforming the continuous attribute values $b_j, b_{1j}^{(m)}, b_{2j}^{(m)}, \cdots, b_{\mathcal{N}_{m,j}}^{(m)}$ into the discrete attribute values $c_j, c_{1j}^{(m)}, c_{2j}^{(m)}, \cdots, c_{\mathcal{N}_{m,j}}^{(m)}$. This form of discretization-based NBC is called dis-NBC in this study.

## 3. Mixed-Attribute Fusion-Based Naive Bayesian Classifier

As stated in the introduction, continuous attribute discretization and the attribute independence assumption limit the generalization performance of the NBC for the continuous attribute classification problem. To cope with these issues, recent studies either introduced complex structures to represent attribute dependencies or applied discretization techniques to transform mixed-attribute values into discrete attribute values [18–20]. This section presents a novel solution to the NBC-based mixed-attribute classification problem by considering attribute dependence and mixed-attribute transformation simultaneously. A mixed-attribute fusion strategy is designed to construct an NBC that can be trained based on the transformed continuous attributes with the minimum attribute dependence.

For discrete attributes of an original mixed-attribute (OMA) data set $\mathbb{D}$, the one-hot encoding technique [21] is applied to transform them into 0-1 numerical attributes; i.e., $a_{ni}^{(m)}, i = 1, 2, \cdots, \mathcal{D}_1$ is encoded as $\left(e_{n1}^{(mi)}, \cdots, e_{n\mathcal{K}_i}^{(mi)}\right)$ for the discrete attribute value of the $n$-th sample $x_n^{(mi)}, n = 1, 2, \cdots, \mathcal{N}_m$, where the following is the case:

$$e_{nk}^{(m)} = \begin{cases} 1, & \text{if } a_{ni}^{(m)} = A_k^{(i)} \\ 0, & \text{if } a_{ni}^{(m)} \neq A_k^{(i)} \end{cases} \tag{9}$$

and $\left\{A_1^{(i)}, A_2^{(i)}, \cdots, A_{\mathcal{K}_i}^{(i)}\right\}$ is $\mathcal{K}_i$ categorical values corresponding to the $i$-th discrete attribute. Then, the one-hot encoded form of the original sample $x_n^{(m)}$ can be expressed as follows.

$$\bar{x}_n^{(m)} = \left(e_{n1}^{(m1)}, \cdots, e_{n\mathcal{K}_1}^{(m1)}, \cdots, e_{n1}^{(m\mathcal{D}_1)}, \cdots, e_{n\mathcal{K}_{\mathcal{D}_1}}^{(m\mathcal{D}_1)}, b_{n1}^{(m)}, \cdots, b_{n\mathcal{D}_2}^{(m)}\right) \quad . \tag{10}$$

In fact, the one-hot encoding technique causes attribute redundancy when extending a discrete attribute into multiple 0-1 numerical attributes. For example, the essence of the following transformation

$$
\begin{bmatrix} A_1 \\ A_1 \\ A_2 \end{bmatrix} \rightarrow \begin{bmatrix} 1 & 0 \\ 1 & 0 \\ 0 & 1 \end{bmatrix}
\tag{11}
$$

is to represent a discrete attribute using two "repetitive" numerical attributes. The main role of one-hot encoding is to transform the discrete attribute values into numbers. The attribute independence assumption is not alleviated when constructing an NBC based on a one-hot encoded attribute (OHEA) data set.

$$
\overline{X} = \begin{bmatrix} \overline{x}_1^{(1)} & \cdots & \overline{x}_{\mathcal{N}_1}^{(1)} & \overline{x}_1^{(2)} & \cdots & \overline{x}_{\mathcal{N}_2}^{(2)} & \cdots & \overline{x}_1^{(\mathcal{M})} & \cdots & \overline{x}_{\mathcal{N}_\mathcal{M}}^{(\mathcal{M})} \end{bmatrix}^{\mathrm{T}}.
\tag{12}
$$

Thus, an autoencoder neural network (ANN) is meticulously designed to solve the problems of attribute redundancy and attribute dependence mentioned above.

An ANN is a special single hidden-layer feed-forward neural network that has the same input matrix and output matrix. The main purpose of training an ANN is to find the optimal input layer weight matrix $\alpha = (\alpha_{rl})_{\mathcal{R} \times \mathcal{L}} = \begin{bmatrix} \alpha_1 & \alpha_2 & \cdots & \alpha_\mathcal{L} \end{bmatrix}$ and output layer matrix $\beta = (\beta_{lr})_{\mathcal{L} \times \mathcal{R}}$ so that the practical output matrix $\overline{X}'$ approximates the true output matrix $\overline{X}$ as closely as possible, where $\mathcal{R} = \sum_{i=1}^{\mathcal{D}_1} \mathcal{K}_i + \mathcal{D}_2$ is the number of input layer nodes of the ANN and $\mathcal{L}$ is the number of hidden layer nodes of the ANN.

The following objective function for ANN is designed to transform one-hot encoded attributes into independent encoded attributes:

$$
\mathrm{L}(\overline{X}, \alpha, \beta) = \lambda \mathrm{L}_1(\overline{X}, \alpha, \beta) + (1 - \lambda) L_2(\overline{X}, \alpha, \beta),
\tag{13}
$$

where $\lambda \in (0, 1)$ is a balance factor. Optimal weight matrices $\overline{\alpha}$ and $\overline{\beta}$ are determined by minimizing the objective function as follows:

$$
\overline{\alpha}, \overline{\beta} = \operatorname*{argmax}_{\substack{\alpha_{rl}, \beta_{lr} \in \Re \\ r=1,2,\cdots,\mathcal{R}; l=1,2,\cdots,\mathcal{L}}} \left\{ \mathrm{L}(\overline{X}, \alpha, \beta) \right\}.
\tag{14}
$$

Here, it is worthwhile to note that the original intention of using an autoencoder neural network is to fuse the mixed attributes and to relax the attribute independence assumption rather than exploring the usage of deep learning technology. When a shallow learning can already meet the requirement of good NBC construction, it is unnecessary to resort to complex and time-consuming deep learning. The excessive attention on attribute transformation with deep learning is out of the scope of this study. Interested readers can refer to specialized studies on the combination of deep learning and supervised learning for more details on such approaches, e.g., deep support vector machine [22], deep decision tree [23], and deep nearest neighbor [24].

The first term of the objective function is the encoding error, which is used to measure the error between the practical output matrix $\overline{X}'$ and the true output matrix $\overline{X}$. It is defined as follows:

$$
\mathrm{L}_1(\overline{X}, \alpha, \beta) = \left\| \overline{X}' - \overline{X} \right\|_2^2 = \left\| \left[ \operatorname{sigmoid}(\overline{X}\alpha) \right] \beta - \overline{X} \right\|_2^2
\tag{15}
$$

where

$$
\overline{\overline{X}} = \operatorname{sigmoid}(\overline{X}\alpha) = \begin{bmatrix} \overline{\overline{x}}_1^{(1)} & \cdots & \overline{\overline{x}}_{\mathcal{N}_1}^{(1)} \cdots & \overline{\overline{x}}_1^{(\mathcal{M})} & \cdots & \overline{\overline{x}}_{\mathcal{N}_\mathcal{M}}^{(\mathcal{M})} \end{bmatrix}^{\mathrm{T}}
\tag{16}
$$

is a $\mathcal{N} \times \mathcal{L}$ hidden-layer output matrix that is a class-independent encoded-attribute (IEA) data set, such that $\overline{\overline{\mathbf{x}}}_n^{(m)} = \left(\overline{\overline{x}}_{n1}^{(m)}, \cdots, \overline{\overline{x}}_{n\mathcal{L}}^{(m)}\right) = \left(\text{sigmoid}\left(\overline{x}_n^{(m)}\alpha_1\right), \cdots, \text{sigmoid}\left(\overline{x}_n^{(m)}\alpha_\mathcal{L}\right)\right)$ is the independent encoded form of the original sample $\mathbf{x}_n^{(m)}$ and

$$\text{sigmoid}(s) = \frac{1}{1 + \exp(-s)}, s \in (-\infty, +\infty) \tag{17}$$

is the sigmoid activation function. To minimize attribute dependence, the attribute dependence term in the objective function is defined as follows:

$$L_2\left(\overline{X}, \alpha, \beta\right) = \frac{2}{\mathcal{L}(\mathcal{L}-1)} \sum_{i=1}^{\mathcal{L}} \sum_{\substack{j=1 \\ j \neq i}}^{\mathcal{L}} I(h_i, h_j), \tag{18}$$

where

$$h_i = \left(\overline{\overline{x}}_{1i}^{(1)}, \cdots, \overline{\overline{x}}_{\mathcal{N}_1,i}^{(1)}, \cdots, \overline{\overline{x}}_{1i}^{(\mathcal{M})}, \cdots, \overline{\overline{x}}_{\mathcal{N}_\mathcal{M},i}^{(\mathcal{M})}\right)^{\text{T}} \tag{19}$$

is the $i$-th independent encoded attribute, and $I(h_i, h_j)$ is the mutual information between independent encoded attributes $h_i$ and $h_j$ and $i, j \in \{1, 2, \cdots, \mathcal{L}\}$ and $i \neq j$.

   The updating rules of $\alpha_{rl}$ and $\beta_{lr}$ are derived as follows. The second term of the objective function is unrelated to the output-layer weights. Thus, the updating rule mainly depends on the encoding error. The gradient descent method is used to determine the updating rule of $\beta_{lr}$. The partial derivative of $L\left(\overline{X}, \alpha, \beta\right)$ with respect to $\beta_{lr}$ is calculated as follows:

$$\Delta\beta_{lr} = \frac{\partial L\left(\overline{X}, \alpha, \beta\right)}{\partial \beta_{lr}} = \lambda \frac{\partial L_1\left(\overline{X}, \alpha, \beta\right)}{\partial \beta_{lr}} \tag{20}$$

and then the updating rule of $\beta_{lr}$ is given as follows

$$\beta_{lr} \leftarrow \beta_{lr} - \xi \times \Delta\beta_{lr}, \tag{21}$$

where $\xi > 0$ is the learning rate. For the input-layer weight $\alpha_{rl}$, the updating rule cannot be derived by using the gradient descent method because of the existence of mutual information terms. Here, a new updating strategy based on the Monte Carlo method [25] is designed as follows:

$$\alpha_{rl} \leftarrow \alpha_{rl} - \zeta \times \Delta\alpha_{rl}, \tag{22}$$

where $\zeta > 0$ is the learning rate and

$$\Delta\alpha_{rl} = \frac{1}{\mathcal{N}} \sum_{m=1}^{\mathcal{M}} \sum_{n=1}^{\mathcal{N}_m} L\left(\overline{x}_n^{(m)}, \alpha, \beta\right) \tag{23}$$

is the approximation of the gradient $\frac{\partial L\left(\overline{X}, \alpha, \beta\right)}{\partial \alpha_{rl}}$.

   Based on the IEA data set, mixed-attribute fusion-based NBC (MAF-NBC) determines the class label for a given new sample $\mathbf{x} = \left(a_1, \cdots, a_{\mathcal{D}_1}, b_1, \cdots, b_{\mathcal{D}_2}\right)$ as follows. First, the one-hot encoded form of $\mathbf{x}$ is expressed as follows:

$$\overline{\mathbf{x}} = \left(e_1^{(1)}, \cdots, e_{\mathcal{K}_1}^{(1)}, \cdots, e_1^{(\mathcal{D}_1)}, \cdots, e_{\mathcal{K}_{\mathcal{D}_1}}^{(\mathcal{D}_1)}, b_1, \cdots, b_{\mathcal{D}_2}\right), \tag{24}$$

where the following.

$$e_k^{(i)} = \begin{cases} 1, & \text{if } a_i = A_k^{(i)} \\ 0, & \text{if } a_i \neq A_k^{(i)} \end{cases}, k = 1, 2, \cdots, \mathcal{K}_i, i = 1, 2, \cdots, \mathcal{D}_1. \tag{25}$$

Second, $\overline{x}$ is transformed into the following independent encoded expression:

$$\overline{\overline{x}} = \left(\overline{\overline{x}}_1, \overline{\overline{x}}_2, \cdots, \overline{\overline{x}}_{\mathcal{L}}\right) = \left(\text{sigmoid}(\overline{x}\alpha_1), \text{sigmoid}(\overline{x}\alpha_2), \cdots, \text{sigmoid}(\overline{x}\alpha_{\mathcal{L}})\right) \quad (26)$$

based on the trained ANN with input weight matrix $\alpha = (\alpha_{rl})_{\mathcal{R} \times \mathcal{L}}$. Third, the class label of x is determined according to Equation (1), where the conditional probability is calculated as follows:

$$
\begin{aligned}
\mathrm{P}(\mathrm{x}|w_m) &= \mathrm{P}(\overline{x}|w_m) = \mathrm{P}(\overline{\overline{x}}|w_m) \\
&= \prod_{l=1}^{\mathcal{L}} \mathrm{P}(\overline{\overline{x}}_l|w_m) = \prod_{l=1}^{\mathcal{L}} \int_{-\infty}^{\overline{\overline{x}}_l} \mathrm{p}(s|w_m)'
\end{aligned}
\quad (27)
$$

where $\mathrm{p}(s|w_m)$ is approximated with a normal *p.d.f.*

$$
f_l^{(m)}(s) = \frac{1}{\sqrt{2\pi}\sigma_l^{(m)}} \exp\left[-\frac{1}{2}\left(\frac{s - \mu_l^{(m)}}{\sigma_l^{(m)}}\right)^2\right],
$$

$$
s \in (-\infty, +\infty), m = 1, 2, \cdots, \mathcal{M}, l = 1, 2, \cdots, \mathcal{L}
\quad (28)
$$

with the mean value

$$\mu_l^{(m)} = \frac{1}{\mathcal{N}_m} \sum_{n=1}^{\mathcal{N}_m} \overline{\overline{x}}_{nl}^{(m)} \quad (29)$$

and standard deviation (std).

$$\sigma_l^{(m)} = \sqrt{\frac{1}{\mathcal{N}_m - 1} \sum_{n=1}^{\mathcal{N}_m} \left[\overline{\overline{x}}_{nl}^{(m)} - \mu_l^{(m)}\right]^2}. \quad (30)$$

Here, an in-depth discussion regarding the normal *p.d.f.* $f_l^{(m)}(\bullet)$ is given. The sigmoid activation function is used in the designed ANN; thus, the outputs corresponding to each hidden layer's nodes obey a quasi-normal probability distribution. A visual comparison of the sigmoid activation function and normal probability distribution functions with standard deviations 0.1, 0.5, 1.0, and 2.0, as shown in Figure 1, clearly demonstrates this empirical conclusion.

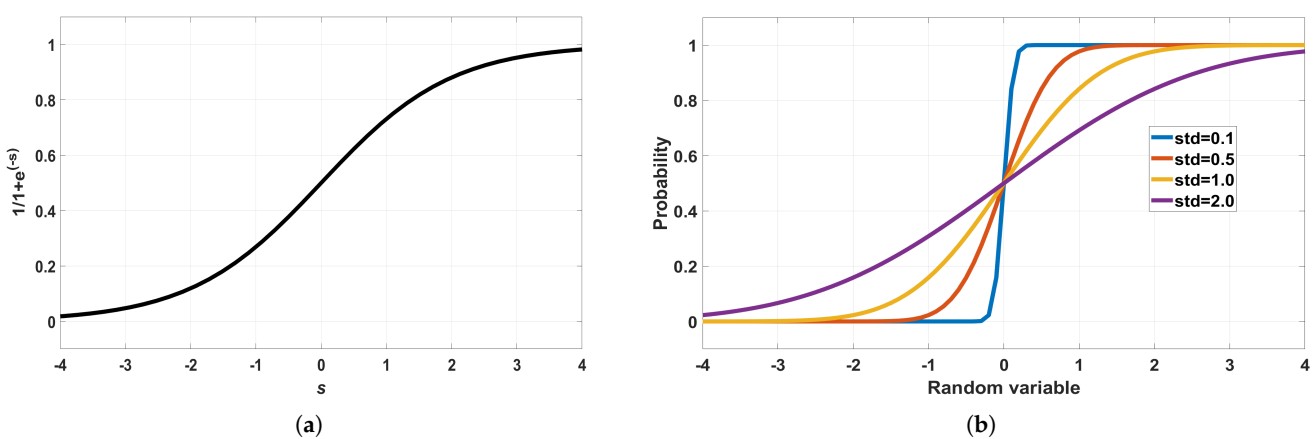

(**a**)  (**b**)

**Figure 1.** Graphical comparison between sigmoid activation and normal distribution. (**a**) Sigmoid activation function. (**b**) Normal probability distribution functions.

For independent attributes, the joint probability distribution is the normal probability distribution if marginal probability distributions are normal probability distributions. Thus, the joint probability $\mathrm{P}(\overline{\overline{x}}|w_m)$ can be modeled as the product of multiple marginal probabilities $\mathrm{P}(\overline{\overline{x}}_l|w_m)$, $l = 1, 2, \cdots, \mathcal{L}$. For the sake of simplicity, hidden-layer biases are not used in the constructed ANN. The role of hidden-layer biases is to control the



bias between the hidden layer's input and the original point of the sigmoid function. When conducting classification or regression tasks, hidden-layer biases are helpful for the generation of high-performance learners. In MAF-NBC, the ANN is designed to transform one-hot encoded attributes into independent encoded attributes that are expected to have minimum dependence. The reasonable objective function shown in Equation (11) is able to guarantee the generation of independent encoded attributes even though hidden-layer biases are not used in ANN. The following experiments will support the aforementioned discussion and conclusion.

## 4. Experimental Settings and Results

A series of experiments are conducted in this section to validate the feasibility, rationality, and effectiveness of the proposed mixed-attribute fusion-based naive Bayesian classifier (MAF-NBC). Experiments were conducted using 20 KEEL [26] mixed-attribute data sets. Their characteristics are listed in Table 1, where the Arabic numerals in parentheses represent the numbers of avaliable categorical values corresponding to the discrete attributes. The data processing strategy proposed by Helal and Otero [27] was used to generate the mixed attributes for the data sets marked without '*' in Table 1. Then, the equal width discretization method was used to transform the continuous attributes into discrete attributes. All data sets as shown in Table 1 can be downloaded from our BaiDuPan or GitHub online storage space. The experiments were run on a PC equipped with an Intel(R) Quadcore 3.00 GHz i5-9400 CPU and 16 GB of RAM.

**Table 1.** Details of 20 mixed-attribute data sets.

| Data sets | Abalone_3_4 | Abalone_9_10 | Adult * | Band * |
|---|---|---|---|---|
| Number of samples | 423 | 1073 | 5000 | 540 |
| Number of continuous attributes | 6 | 6 | 9 | 18 |
| Number of discrete attributes | 2 (2, 4) | 2 (2, 4) | 5 (5, 7, 5, 4, 2) | 5 (3, 5, 2, 3, 3) |
| Number of classes | 2 | 2 | 2 | 2 |
| data sets | Bd | Bp | Heart | Ionosphere |
| Number of samples | 569 | 198 | 270 | 351 |
| Number of continuous attributes | 23 | 25 | 9 | 25 |
| Number of discrete attributes | 7 (4, 6, 5, 3, 2, 4, 3) | 8 (6, 3, 5, 4, 5, 3, 4, 2) | 4 (4, 3, 3, 4) | 7 (4, 6, 5, 3, 4, 3, 5) |
| Number of classes | 2 | 2 | 2 | 2 |
| Data sets | Libras | Page blocks | Parkinsons | Ring |
| Number of samples | 360 | 547 | 195 | 500 |
| Number of continuous attributes | 85 | 7 | 17 | 15 |
| Number of discrete attributes | 15 (4, 5, 7, 5, 8, 4, 5, 6, 4, 5, 6, 7, 3, 4, 5) | 3 (3, 5, 4) | 5 (4, 5, 6, 5, 3) | 5 (3, 4, 5, 3, 4) |
| Number of classes | 15 | 5 | 2 | 2 |
| Data sets | Segment | Sonar | SPECTF | Vehicle |
| Number of samples | 2310 | 208 | 267 | 846 |
| Number of continuous attributes | 14 | 50 | 36 | 6 |
| Number of discrete attributes | 5 (3, 6, 5, 4, 3) | 10 (6, 4, 3, 5, 6, 7, 5, 3, 4, 6) | 8 (4, 6, 5, 5, 3, 6, 7, 4) | 2 (4, 3) |
| Number of classes | 7 | 2 | 2 | 4 |
| Data sets | Vowel | Wine | WineQR | WineQW |
| Number of samples | 528 | 178 | 1599 | 489 |
| Number of continuous attributes | 7 | 10 | 8 | 8 |
| Number of discrete attributes | 3 (3, 3, 4) | 3 (4, 5, 4) | 3 (4, 4, 3) | 3 (3, 3, 3) |
| Number of classes | 11 | 3 | 6 | 6 |

### 4.1. Feasibility Validation of MAF-NBC

A first experiment was performed to validate the feasibility of the MAF-NBC method by checking the convergence of ANN weights by the iterative update process. This experiment was conducted on the representative *Vowel* data set, which has three discrete attributes and seven continuous attributes. The experimental results are the average values corresponding to 10 independent ANN training. Ten ANNs with 50 hidden-layer nodes were constructed with random weights in the $[-1, 1]$ range, a balance factor $\lambda = 0.50$, and learning rates $\xi = \zeta = 0.01$. Figure 2 depicts the variation trends of the 1-norms for the input layer and output layer weights as the iteration number increased. Figure 3

presents the variation trends of encoding error and attribute dependence as the iteration number increased.

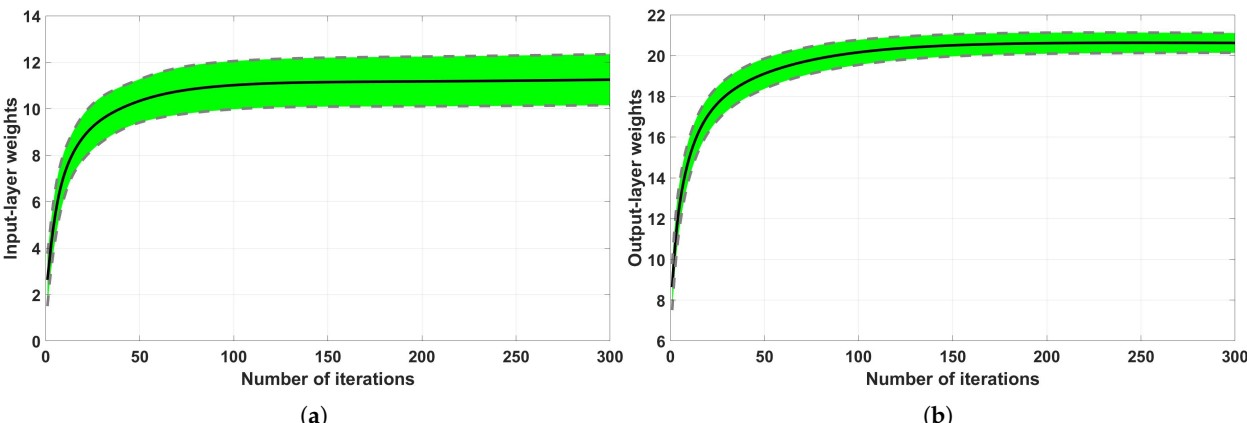

**Figure 2.** Convergence of the ANN's weights. (**a**) Input-layer weights. (**b**) Output-layer weights.

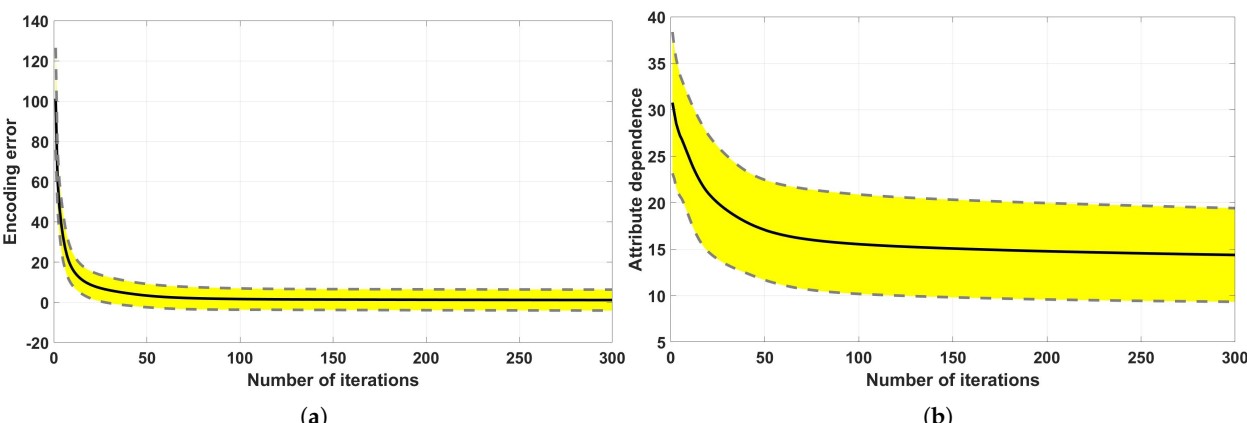

**Figure 3.** Convergence of the ANN's objective function. (**a**) Encoding error. (**b**) Attribute dependence.

Figure 2 shows that the sums of absolute values corresponding to the input layer and output-layer weights become more and more stable with the number of iterations. In addition, Figure 3 shows that the encoding error and attribute dependence decrease gradually and then remain unchanged. These experimental results reveal that the updating rules of Equations (15) and (16) are effective for the determination of optimal ANN weights. In fact, Equation (17) is the approximation of the expected loss $\mathrm{E}_{\overline{\mathcal{X}}}\left[\frac{\partial \mathrm{L}(\overline{\mathcal{X}}, \alpha, \beta)}{\partial \alpha_{rl}}\right]$, where $\overline{\mathcal{X}}$ is the domain of the objective function $\mathrm{L}(\bullet, \alpha, \beta)$. Although the gradient of input layer weights cannot be analytically calculated due to the existence of the attribute dependence term, Equation (17) uses the mean of the objective function values corresponding to all samples in the one-hot encoded attribute data set to approximate the gradient. The convergence of output layer weights guarantees the convergence of input layer weights, because the update of $\alpha_{rl}$ depends on the update of $\beta_{lr}$. This experiment demonstrates the feasibility of transforming the one-hot encoded attributes into independent encoded attributes.

### 4.2. Rationality Validation of MAF-NBC

A second experiment was carried out to evaluate the rationality of MAF-NBC in terms of whether the designed ANN can transform the one-hot encoded attributes corresponding to the original mixed attributes into independent encoded attributes. On the representative *Page_small* data set, an ANN with 50 hidden-layer nodes was constructed with random weights in the $[-1, 1]$ interval, a balance factor $\lambda = 0.50$, and learning rates $\xi = \zeta = 0.01$. Ten representative encoded attributes were selected from the hidden-layer

outputs of ANN. The dependence between the two encoded attributes was measured with the mutual information, which is calculated with *sklearn.metrics.mutual_info_score* (https://scikit-learn.org/stable/modules/generated/sklearn.metrics.mutual_info_score.html, accessed on 14 October 2022) package of the *scikit-learn* machine learning library.

Figure 4 displays a series of heatmaps corresponding to iterations #1, #50, #100, and #150. These heatmaps show the change in attribute dependence as the iteration number increases during the ANN's training. It can be clearly seen that the dependence between encoded attribute gradually decreases as ANN weights are updated. It indicates that independent attributes can be obtained by transforming the original mixed attributes into the encoded attributes. The experimental results shown in Figure 4 are consistent with the experimental results shown in Figure 3b, where the attribute dependence gradually decreases with the increase in iteration number. This experiment demonstrated that the ANN is able to transform the original mixed attributes into independent encoded attributes for NBC construction.

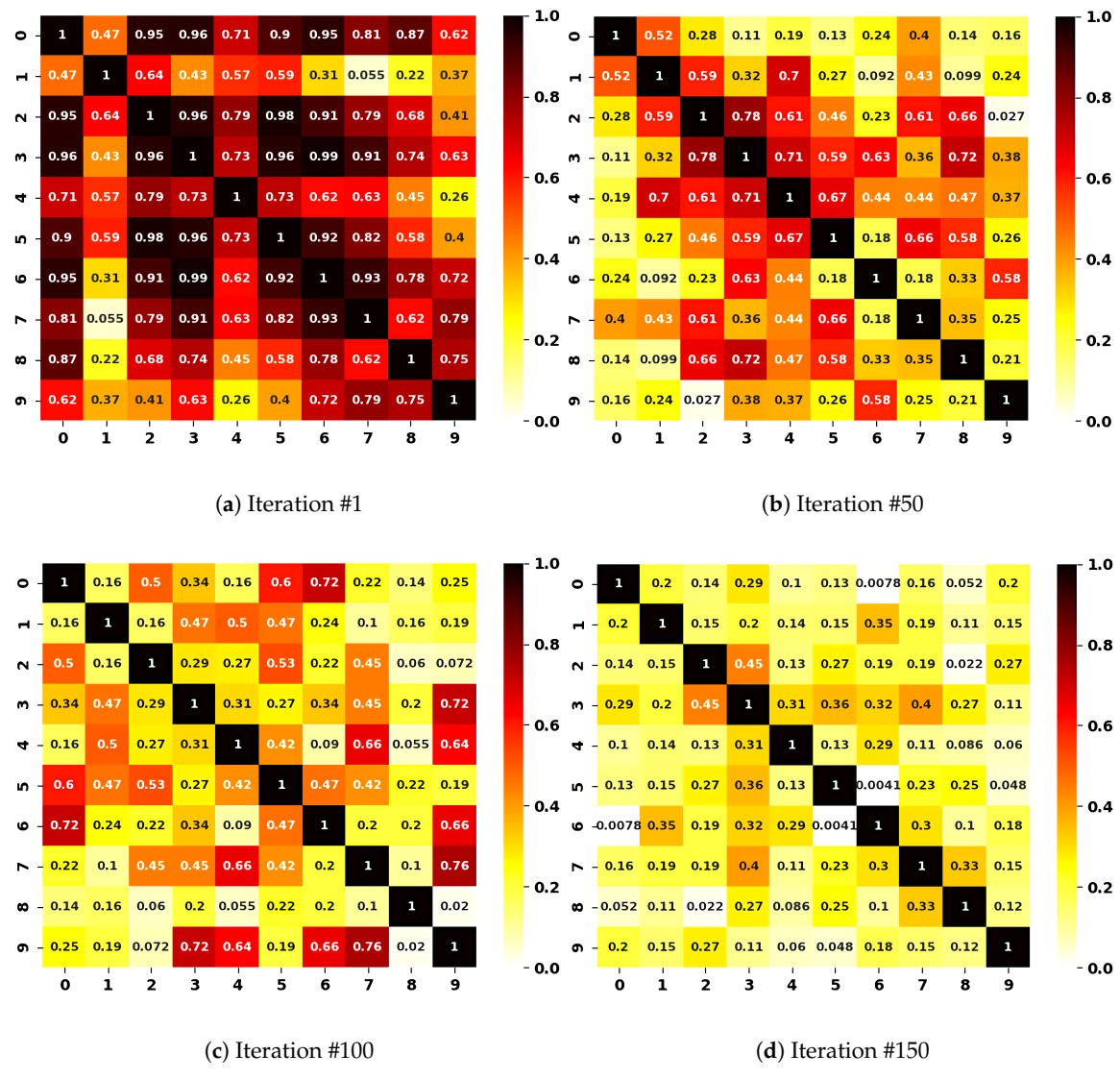

**Figure 4.** Dependence change of the encoded continuous attributes.

*4.3. Effectiveness Validation of MAF-NBC*

A third experiment was performed to evaluate the effectiveness of MAF-NBC. This was performed by comparing the classification performances of MAF-NBC with Dis-NBC, FNBC [3], TAN [4], AODE [5], HNB [6], DFW-NBC [7], CFW-NBC [8], and ICA-NBC [10]. All Bayesian algorithms were implemented using the Python programming language. Twenty KEEL mixed-attribute data sets (described in Table 1) were selected to test the training and testing accuracies of the compared algorithms. For each data set, independent training and testing were conducted 10 times with random data set partitions, i.e., 70% samples to train the different Bayes algorithms and 30% to test their generalization capabilities. All ANNs were constructed with $2\mathcal{L}$ hidden-layer nodes, initialized with random weights in the $[-1, 1]$ range, and setup with the learning parameters $\lambda = 0.50$ and $\xi = \zeta = 0.01$. The mean and standard derivation of the 10 training and testing accuracies are listed for each algorithm in Tables 2 and 3, respectively.

The training and testing performances of MAF-NBC were statistically validated by comparing with eight other Bayes algorithms on the 20 data sets. For the given significance level of 0.05, the critical difference (CD) value [28] is calculated as follows:

$$\text{CD} = 3.102 \times \sqrt{\frac{9 \times (9 + 1)}{6 \times 20}} \approx 2.686, \tag{31}$$

where the number of compared algorithms is nine, and the number of data sets is 20. In Figure 5, an interval of one CD value can be observed to the left and right of the average rank of MAF-NBC. Any algorithm with a rank outside this area is significantly different from MAF-NBC. It is found that the ranks of MAF-NBC corresponding to the training and testing accuracies are obviously smaller than the other algorithms. In addition, the number of wins for MAF-NBC on 20 data sets is at least $\frac{20}{2} + 1.96 \times \frac{\sqrt{20}}{2} \approx 14$ for the significance level of 0.05. Furthermore, it can be said that MAF-NBC has significantly improved testing accuracies than Dis-NBC (20 wins), FNBC (20 wins), TAN (20 wins), AODE (20 wins), HNB (20 wins), DFW-NBC (19 wins), CFW-NBC (19 wins), and ICA-NBC (18 wins) under a significance level of 0.10. This indicates that MAF-NBC is significantly better for classification than the other algorithms on the selected data sets. The experimental results and statistical analyses demonstrate the effectiveness of MAF-NBC and indicate that MAF-NBC is a viable method to handle mixed-attribute classification tasks. MAF-NBC does not modify the simple model structure of the NBC and preserves the amount of information of the original mixed-attribute data set as much as possible, because the ANN transforms the original mixed attributes into encoded attributes rather than selecting or extracting independent attributes from the original mixed attributes.

**Table 2.** Comparison of the training accuracies of MAF-NBC, Dis-NBC, FNB, TAN, AODE, HNB, DFW-NBC, CFW-NBC, and ICA-NBC.

| Data Set | MAF-NBC | Dis-NBC | FNBC | TAN | AODE | HNB | DFW-NBC | CFW-NBC | ICA-NBC |
|---|---|---|---|---|---|---|---|---|---|
| Abalone_3_4 | **0.803** ± 0.036 | 0.721 ± 0.032 | 0.736 ± 0.026 | 0.767 ± 0.038 | 0.755 ± 0.032 | 0.744 ± 0.029 | 0.747 ± 0.018 | 0.755 ± 0.017 | 0.781 ± 0.045 |
| Abalone_9_10 | 0.607 ± 0.029 | 0.597 ± 0.012 | 0.592 ± 0.021 | 0.595 ± 0.025 | 0.596 ± 0.024 | 0.587 ± 0.041 | 0.617 ± 0.008 | 0.605 ± 0.014 | **0.619** ± 0.017 |
| Adult | **0.870** ± 0.013 | 0.782 ± 0.014 | 0.792 ± 0.015 | 0.807 ± 0.013 | 0.829 ± 0.008 | 0.811 ± 0.017 | 0.805 ± 0.011 | 0.813 ± 0.006 | 0.851 ± 0.015 |
| Band | **0.718** ± 0.027 | 0.669 ± 0.028 | 0.637 ± 0.034 | 0.671 ± 0.035 | 0.663 ± 0.024 | 0.681 ± 0.028 | 0.703 ± 0.015 | **0.718** ± 0.017 | 0.714 ± 0.016 |
| Bd | **0.991** ± 0.002 | 0.919 ± 0.004 | 0.938 ± 0.010 | 0.961 ± 0.009 | 0.959 ± 0.008 | 0.953 ± 0.011 | 0.959 ± 0.007 | 0.970 ± 0.004 | 0.983 ± 0.007 |
| Bp | **0.852** ± 0.022 | 0.788 ± 0.023 | 0.793 ± 0.027 | 0.803 ± 0.029 | 0.801 ± 0.034 | 0.791 ± 0.041 | 0.841 ± 0.036 | 0.824 ± 0.033 | 0.837 ± 0.027 |
| Heart | **0.924** ± 0.014 | 0.699 ± 0.013 | 0.824 ± 0.017 | 0.858 ± 0.015 | 0.849 ± 0.021 | 0.867 ± 0.029 | 0.886 ± 0.013 | 0.905 ± 0.015 | 0.901 ± 0.017 |
| Ionosphere | **0.981** ± 0.005 | 0.901 ± 0.006 | 0.917 ± 0.011 | 0.934 ± 0.009 | 0.932 ± 0.013 | 0.926 ± 0.021 | 0.928 ± 0.012 | 0.941 ± 0.015 | 0.974 ± 0.004 |
| Libras | 0.905 ± 0.007 | **0.938** ± 0.015 | 0.857 ± 0.019 | 0.899 ± 0.015 | 0.847 ± 0.016 | 0.864 ± 0.015 | 0.911 ± 0.016 | 0.910 ± 0.015 | 0.908 ± 0.011 |
| Page_small | **0.952** ± 0.010 | 0.915 ± 0.005 | 0.911 ± 0.007 | 0.909 ± 0.018 | 0.919 ± 0.014 | 0.914 ± 0.017 | 0.931 ± 0.006 | 0.933 ± 0.008 | 0.937 ± 0.007 |
| Parkinsons | **0.872** ± 0.029 | 0.782 ± 0.021 | 0.801 ± 0.022 | 0.808 ± 0.025 | 0.814 ± 0.027 | 0.821 ± 0.021 | 0.827 ± 0.022 | 0.841 ± 0.025 | 0.852 ± 0.024 |
| Ring | 0.937 ± 0.015 | 0.801 ± 0.017 | 0.836 ± 0.015 | 0.855 ± 0.028 | 0.869 ± 0.024 | 0.858 ± 0.027 | 0.916 ± 0.010 | 0.948 ± 0.006 | **0.956** ± 0.008 |
| Segment | 0.940 ± 0.007 | 0.878 ± 0.010 | 0.889 ± 0.019 | 0.897 ± 0.016 | 0.895 ± 0.014 | 0.901 ± 0.016 | 0.928 ± 0.006 | 0.945 ± 0.009 | **0.959** ± 0.005 |
| Sonar | **0.865** ± 0.019 | 0.823 ± 0.029 | 0.834 ± 0.027 | 0.839 ± 0.025 | 0.844 ± 0.018 | 0.840 ± 0.025 | 0.841 ± 0.027 | 0.847 ± 0.029 | 0.851 ± 0.024 |
| Spectf | **0.908** ± 0.011 | 0.768 ± 0.023 | 0.771 ± 0.017 | 0.783 ± 0.019 | 0.779 ± 0.015 | 0.756 ± 0.029 | 0.779 ± 0.016 | 0.789 ± 0.014 | 0.883 ± 0.019 |
| Vehicle | **0.771** ± 0.016 | 0.537 ± 0.005 | 0.544 ± 0.011 | 0.576 ± 0.016 | 0.589 ± 0.019 | 0.567 ± 0.026 | 0.604 ± 0.022 | 0.611 ± 0.021 | 0.705 ± 0.017 |
| Vowel | **0.869** ± 0.017 | 0.829 ± 0.021 | 0.839 ± 0.017 | 0.836 ± 0.019 | 0.840 ± 0.016 | 0.847 ± 0.019 | 0.843 ± 0.021 | 0.848 ± 0.016 | 0.833 ± 0.023 |
| Wine | **0.999** ± 0.001 | 0.963 ± 0.021 | 0.967 ± 0.017 | 0.975 ± 0.006 | 0.981 ± 0.004 | 0.968 ± 0.007 | **0.999** ± 0.001 | **0.999** ± 0.001 | 0.968 ± 0.019 |
| WineQR | **0.738** ± 0.023 | 0.563 ± 0.014 | 0.578 ± 0.034 | 0.580 ± 0.033 | 0.591 ± 0.031 | 0.602 ± 0.037 | 0.638 ± 0.020 | 0.717 ± 0.019 | 0.688 ± 0.036 |
| WineQW | **0.584** ± 0.028 | 0.487 ± 0.014 | 0.524 ± 0.016 | 0.537 ± 0.024 | 0.533 ± 0.016 | 0.541 ± 0.021 | 0.551 ± 0.028 | 0.561 ± 0.025 | 0.577 ± 0.021 |
| **Average** | **0.854** ± 0.017 | 0.768 ± 0.016 | 0.779 ± 0.019 | 0.795 ± 0.021 | 0.794 ± 0.016 | 0.792 ± 0.024 | 0.813 ± 0.016 | 0.824 ± 0.015 | 0.839 ± 0.018 |

**Table 3.** Comparison of the testing accuracies of MAF-NBC, Dis-NBC, FNB, TAN, AODE, HNB, DFW-NBC, CFW-NBC, and ICA-NBC.

| Data Set | MAF-NBC | Dis-NBC | FNBC | TAN | AODE | HNB | DFW-NBC | CFW-NBC | ICA-NBC |
|---|---|---|---|---|---|---|---|---|---|
| Abalone_3_4 | **0.759** ± 0.030 | 0.685 ± 0.042 | 0.701 ± 0.049 | 0.742 ± 0.032 | 0.733 ± 0.037 | 0.728 ± 0.029 | 0.723 ± 0.031 | 0.751 ± 0.027 | 0.755 ± 0.043 |
| Abalone_9_10 | **0.603** ± 0.028 | 0.587 ± 0.024 | 0.588 ± 0.026 | 0.587 ± 0.027 | 0.589 ± 0.028 | 0.581 ± 0.030 | 0.589 ± 0.031 | 0.582 ± 0.031 | 0.602 ± 0.032 |
| Adult | **0.842** ± 0.014 | 0.795 ± 0.011 | 0.788 ± 0.019 | 0.801 ± 0.025 | 0.817 ± 0.012 | 0.807 ± 0.027 | 0.804 ± 0.012 | 0.811 ± 0.014 | 0.824 ± 0.009 |
| Band | **0.647** ± 0.045 | 0.606 ± 0.028 | 0.601 ± 0.028 | 0.626 ± 0.036 | 0.618 ± 0.038 | 0.609 ± 0.031 | 0.619 ± 0.034 | 0.630 ± 0.036 | 0.619 ± 0.041 |
| Bd | **0.973** ± 0.011 | 0.915 ± 0.017 | 0.921 ± 0.013 | 0.956 ± 0.012 | 0.950 ± 0.010 | 0.946 ± 0.014 | 0.933 ± 0.015 | 0.944 ± 0.016 | 0.948 ± 0.014 |
| Bp | **0.764** ± 0.040 | 0.739 ± 0.047 | 0.746 ± 0.039 | 0.740 ± 0.057 | 0.739 ± 0.052 | 0.737 ± 0.065 | 0.755 ± 0.036 | 0.747 ± 0.041 | 0.746 ± 0.049 |
| Heart | **0.876** ± 0.028 | 0.703 ± 0.022 | 0.711 ± 0.025 | 0.739 ± 0.037 | 0.728 ± 0.034 | 0.789 ± 0.039 | 0.859 ± 0.032 | 0.864 ± 0.035 | 0.851 ± 0.038 |
| Ionosphere | **0.947** ± 0.027 | 0.894 ± 0.024 | 0.906 ± 0.021 | 0.917 ± 0.017 | 0.907 ± 0.025 | 0.901 ± 0.027 | 0.923 ± 0.029 | 0.927 ± 0.020 | 0.931 ± 0.026 |
| Libras | **0.778** ± 0.039 | 0.743 ± 0.047 | 0.756 ± 0.058 | 0.751 ± 0.021 | 0.749 ± 0.048 | 0.748 ± 0.039 | 0.759 ± 0.036 | 0.764 ± 0.039 | 0.759 ± 0.038 |
| Page_small | 0.901 ± 0.019 | 0.896 ± 0.015 | 0.889 ± 0.013 | 0.891 ± 0.016 | 0.881 ± 0.019 | 0.898 ± 0.017 | 0.912 ± 0.015 | **0.913** ± 0.011 | 0.899 ± 0.012 |
| Parkinsons | **0.826** ± 0.042 | 0.756 ± 0.043 | 0.769 ± 0.049 | 0.779 ± 0.059 | 0.771 ± 0.047 | 0.782 ± 0.039 | 0.786 ± 0.034 | 0.807 ± 0.044 | 0.779 ± 0.030 |
| Ring | **0.928** ± 0.017 | 0.781 ± 0.041 | 0.831 ± 0.038 | 0.849 ± 0.049 | 0.834 ± 0.041 | 0.879 ± 0.049 | 0.870 ± 0.028 | 0.910 ± 0.011 | 0.925 ± 0.016 |
| Segment | 0.913 ± 0.016 | 0.869 ± 0.019 | 0.874 ± 0.014 | 0.886 ± 0.018 | 0.895 ± 0.016 | 0.889 ± 0.020 | 0.899 ± 0.015 | 0.906 ± 0.013 | **0.922** ± 0.010 |
| Sonar | 0.816 ± 0.036 | 0.758 ± 0.046 | 0.769 ± 0.053 | 0.779 ± 0.038 | 0.806 ± 0.034 | 0.787 ± 0.023 | 0.796 ± 0.036 | 0.801 ± 0.042 | **0.837** ± 0.029 |
| Spectf | **0.823** ± 0.023 | 0.716 ± 0.037 | 0.727 ± 0.027 | 0.778 ± 0.041 | 0.794 ± 0.030 | 0.757 ± 0.041 | 0.748 ± 0.035 | 0.757 ± 0.030 | 0.801 ± 0.029 |
| Vehicle | **0.611** ± 0.029 | 0.483 ± 0.037 | 0.506 ± 0.039 | 0.529 ± 0.032 | 0.520 ± 0.049 | 0.518 ± 0.054 | 0.537 ± 0.029 | 0.549 ± 0.032 | 0.589 ± 0.028 |
| Vowel | **0.859** ± 0.023 | 0.781 ± 0.057 | 0.781 ± 0.052 | 0.799 ± 0.058 | 0.795 ± 0.057 | 0.801 ± 0.033 | 0.811 ± 0.029 | 0.821 ± 0.024 | 0.816 ± 0.027 |
| Wine | **0.961** ± 0.019 | 0.928 ± 0.024 | 0.926 ± 0.029 | 0.937 ± 0.027 | 0.946 ± 0.021 | 0.939 ± 0.026 | 0.958 ± 0.017 | 0.945 ± 0.022 | 0.931 ± 0.016 |
| WineQR | **0.637** ± 0.018 | 0.489 ± 0.025 | 0.509 ± 0.038 | 0.519 ± 0.037 | 0.528 ± 0.034 | 0.524 ± 0.031 | 0.563 ± 0.028 | 0.587 ± 0.030 | 0.559 ± 0.041 |
| WineQW | **0.534** ± 0.036 | 0.477 ± 0.043 | 0.503 ± 0.033 | 0.508 ± 0.038 | 0.501 ± 0.041 | 0.509 ± 0.029 | 0.508 ± 0.041 | 0.515 ± 0.028 | 0.529 ± 0.039 |
| **Average** | **0.800** ± 0.027 | 0.730 ± 0.032 | 0.740 ± 0.033 | 0.756 ± 0.034 | 0.755 ± 0.034 | 0.756 ± 0.033 | 0.768 ± 0.027 | 0.777 ± 0.027 | 0.781 ± 0.028 |

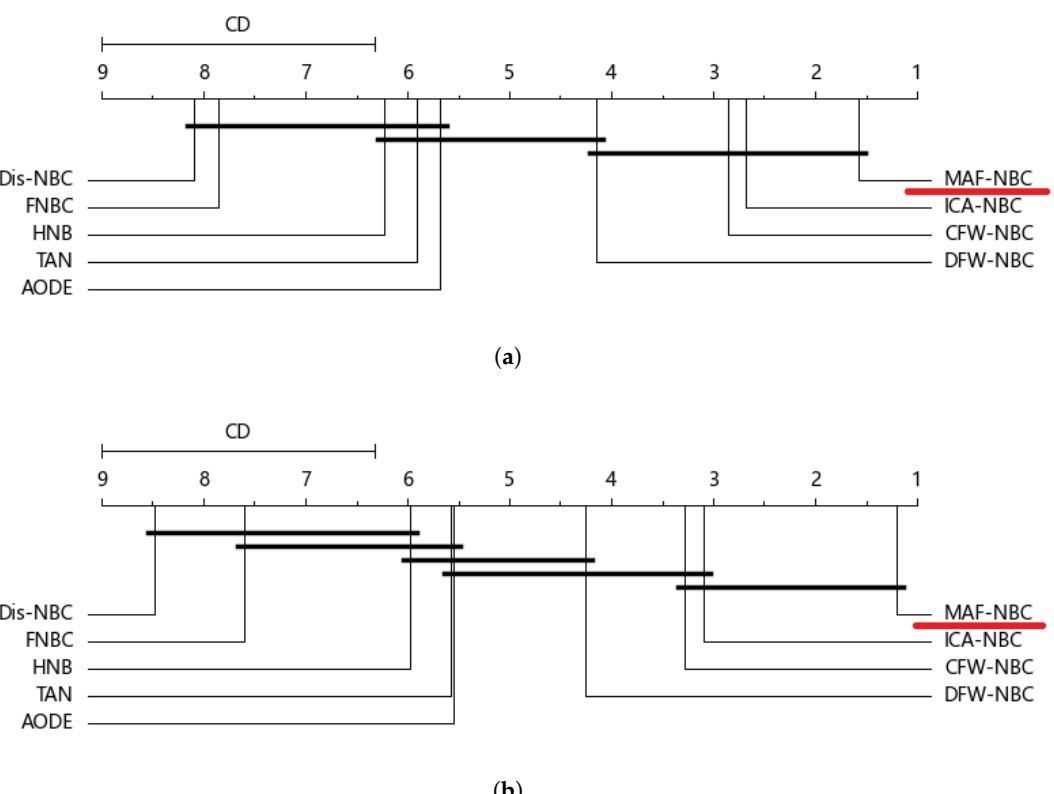

(**a**)

(**b**)

**Figure 5.** CD diagrams corresponding to comparisons in Tables 2 and 3. (**a**) CD diagram of training accuracies. (**b**) CD diagram of testing accuracies.

## 5. Conclusions and Future Works

This paper presented a novel NBC training method for the mixed-attribute data classification problem without continuous attribute discretization and complex Bayesian network structure learning. The original mixed attributes were transformed into a series of continuous attributes with minimum dependence using an autoencoder neural network. To obtain optimal network weights, an effective objective function was designed, and corresponding weight updating rules were derived. The experimental results finally demonstrated improved classification performance for the novel Bayes model in comparison with eight state-of-the-art Bayesian algorithms. The technical advantages of MAF-NBC are four-fold:

- Preserving the information. The autoencoder neural network transforms mixed attributes into independent encoded attributes, which can also be decoded to restore the original mixed attributes. Thus, useful information from the original data set is conserved as much as possible.
- Simple model structure. MAF-NBC effectively handles the attribute dependence problem between discrete attributes and continuous attributes by transforming original attributes into encoded attributes. The simple structure of the NBC was not modified by providing a data transformation-oriented preprocessing strategy.
- Stable training process. The efficient weight updating strategy guarantees the convergence of the autoencoder neural network and thus provided stable data transformation results. In addition, the quasi-normal distribution of independent encoded attributes provided accurate calculations for the joint conditional probability in the NBC.
- Low computation complexity. A single hidden-layer encoder neural network was used to fuse the original mixed attributes. Training to find optimal network weights has a low computational complexity. As mentioned in Section 3, a shallow learning mechanism was able to satisfy the demand of NBC construction. Hence, it is unnec-

essary to explore the availability of deep learning for data transformation-oriented NBC training.

In future studies, we plan (1) to implement MAF-NBC in a distributed environment so that it can be used to deal with large-scale mixed-attribute data-classification problems and (2) to utilize the autoencoder neural network to transform mixed attributes into dependent continuous attributes to construct a non-naive Bayesian classifier [29] based on the joint probability density function estimation technique.

**Author Contributions:** Data curation, formal analysis, and writing—original draft preparation, G.O.; methodology, writing—original draft preparation and review and editing, Y.H.; investigation and writing—review and editing, P.F.-V.; supervision, J.Z.H. All authors have read and agreed to the published version of the manuscript.

**Funding:** This research was funded by the Scientific Research Foundation of National Natural Science Foundation of China (61972261) and Basic Research Foundations of Shenzhen (JCYJ20210324093609026 and JCYJ20200813091134001).

**Institutional Review Board Statement:** None.

**Informed Consent Statement:** None.

**Data Availability Statement:** The data are available in BaiduPan https://pan.baidu.com/s/1741 7zF-IlP6lW_ut7lRTqA (accessed on 14 October 2022) with extraction code *fcju* or GitHub platform https://github.com/ouguiliang110/DataSet_of_MAF-NBC (accessed on 14 October 2022).

**Acknowledgments:** The authors would like to thank the editors and anonymous reviewers who carefully read the paper and provided valuable suggestions that considerably improved the paper. This paper was recommended by the 6th Asian Conference on Artificial Intelligence Technology (ACAIT 2022) which is the 2022 Top Academic Conference recognized by China Association for Science and Technology (CAST).

**Conflicts of Interest:** The authors declare no conflicts of interest.

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
