# Peer review of "A Novel Mixed-Attribute Fusion-Based Naive Bayesian Classifier"

_applsci, doi:10.3390/app122010443_

Round 1

Reviewer 1 Report

In this work, the authors proposed a mixed-attribute fusion-based NBC to make a data set into a series of encoded attributes with maximum independence. The authors have validated the feasibility, rationality, and effectiveness of the method. Their methods show better performance for classification process than several Bayesian-based methods. Here are my comments.

1. The X-axis of Figure 1a and 1b are different. The sigmoid function is similar to the normal probability distribution function with standard error of 0.1. It would be better to use the same scale for comparison.

2. Table 2 and 3 contain too much data, and one may hardly find useful information by looking at these tables. I suggest to use heatmaps to represent the data, and move the Table 2 and 3 to the Supporting Information (If there is such part).  

3. What is the reason of using 20 KEEL mixed attribute data sets? Why not other data sets? The authors used these data sets to assess their methods, and the generalization of these data sets should be well explained.

4. Figure 4 shows the dependence change against different iterations. And  is it done in one training process? The authors may need to conduct several parallel experiments to verify the extent of changes among different tests.

5. I suggest the data is uploaded to the journal website, or uploaded on a commonly accepted platform, such as Github. Users who are not familiar with BaiduPan may have difficulty to access the data.

Author Response

Reviewer #1:

We are very grateful for the invaluable comments/suggestions provided by Reviewer #1,

which have greatly enlightened us regarding the Gaussian process-based random weight network.

In this work, the authors proposed a mixed-attribute fusion-based NBC to make a data set into a series of encoded attributes with maximum independence. The authors have validated the feasibility, rationality, and effectiveness of the method. Their methods show better performance for classification process than several Bayesian-based methods. Here are my comments.

Response: Thank the Respected Reviewer #1 very much for your evaluation and encouragement!

We are very grateful for your through reading and insightful suggestions, which have helped us to significantly improve the paper.

The changes made to address your concerns are detailed below.

  1. The X-axis of Figure 1a and 1b are different. The sigmoid function is similar to the normal probability distribution function with standard error of 0.1. It would be better to use the same scale for comparison.

Response: Thanks a lot for this helpful suggestion!

We have adjusted the scale of Figure 1(a) so that the subfigures in Figure 1 have the same scale. Please check the revised Figure 1 in page 7 or the following screenshot.

  1. Table 2 and 3 contain too much data, and one may hardly find useful information by looking at these tables. I suggest to use heatmaps to represent the data, and move the Table 2 and 3 to the Supporting Information (If there is such part).

Response: Thanks a lot for this helpful suggestion!

We have emphasized the better results in Tables 2 and 3 in boldfaces so that the useful information can be easily and clearly found by our readers. Please check the revised Table 2 in page 11 and Table 3 in page 12.

  1. What is the reason of using 20 KEEL mixed attribute data sets? Why not other data sets? The authors used these data sets to assess their methods, and the generalization of these data sets should be well explained.

Response: This is a good question. Thank you!

MAF-NBC is deliberately designed to deal with the classification problem for mixed attribute data set. Thus, we selected 20 KEEL mixed attribute data sets in our experiments to validate the feasibility, rationality, and effectiveness of proposed MAF-NBC.

  1. Figure 4 shows the dependence change against different iterations. And is it done in one training process? The authors may need to conduct several parallel experiments to verify the extent of changes among different tests.

Response: This is a good question. Thank you!

Yes, the dependence change against different iterations was done in one training process. Figure 4 showed the increase of attribute independence on a representative data set when the mixed attributes were gradually encoded with autoencoder neural network. In fact, we have tried all data sets used in our experiments and selected the page_small data set as an example due to the page limitation.

  1. I suggest the data is uploaded to the journal website, or uploaded on a commonly accepted platform, such as Github. Users who are not familiar with BaiduPan may have difficulty to access the data.

Response: Thanks a lot for this helpful suggestion!

We have uploaded the data sets on GitHub platform:

https://github.com/ouguiliang110/DataSet_of_MAF-NBC.

Thank you again, Reviewer #1, for your thorough review of our submission.

Your invaluable comments and suggestions have helped us improving the manuscript.

We hope that our revisions are satisfactory.

Reviewer 2 Report

Actually, regarding section 4.3 (Effectiveness validation of MAF-NBC), the authors have compared the proposed algorithm with out-of-date algorithms. Thus, the author must add more recent algorithms to enhance this section.

Author Response

Reviewer #2:

We are very grateful for the invaluable comments/suggestions provided by Reviewer #2,

which have greatly enlightened us regarding the Gaussian process-based random weight network.

Actually, regarding section 4.3 (Effectiveness validation of MAF-NBC), the authors have compared the proposed algorithm with out-of-date algorithms. Thus, the author must add more recent algorithms to enhance this section.

Response: Thank the Respected Reviewer #2 very much for your evaluation and encouragement!

The recent NBC variant abbreviated as CFW-NBC [B1] has been used in our experiment to compare its training and testing accuracies with our proposed MAF-NBC.

[B1] Jiang L., Zhang L., Li C., et al.: A correlation-based feature weighting filter for Naive Bayes. IEEE Transactions on Knowledge and Data Engineering, 31(2): 201-213 (2018)

Thanks again to Reviewer #2 for thorough reviewing our submission.

Your invaluable comments and suggestions have been very helpful to improve the manuscript.

We hope that our revisions are satisfactory.
